# Characterization of the Barley Net Blotch Pathosystem at the Center of Origin of Host and Pathogen

**DOI:** 10.3390/pathogens8040275

**Published:** 2019-11-29

**Authors:** Moshe Ronen, Hanan Sela, Eyal Fridman, Rafael Perl-Treves, Doris Kopahnke, Alexandre Moreau, Roi Ben-David, Arye Harel

**Affiliations:** 1Department of vegetables and field crops, Institute of Plant Science, Agricultural Research Organization, The Volcani Center, Rishon LeZion 7505101, Israel; ronen@volcani.agri.gov.il (M.R.); fridmane@volcani.agri.gov.il (E.F.); moreaualexandre@gmail.com (A.M.); 2The Mina & Everard Goodman Faculty of Life Sciences, Bar-Ilan University, Ramat-Gan 5290002, Israel; perl@mail.biu.ac.il; 3Institute for Cereal Crops Improvement, Tel Aviv University, Tel Aviv 6997801, Israel; hans@tauex.tau.ac.il; 4Federal Research Institute for Cultivated Plants, Institute for Resistance Research and Stress Tolerance, Julius Kühn-Institut (JKI), Erwin-Baur-Str. 27, 06484 Quedlinburg, Germany; doris.kopahnke@julius-kuehn.de

**Keywords:** barley, net blotch (NB), *Pyrenophora teres* f. *teres* (*Ptt*), *Pyrenophora teres* f. *maculata* (*Ptm*), resistance, *Hordeum vulgare* ssp. *spontaneum*

## Abstract

Net blotch (NB) is a major disease of barley caused by the fungus *Pyrenophora teres* f. *teres* (*Ptt*), and *P. teres* f. *maculata* (*Ptm*). *Ptt* and *Ptm* infect the cultivated crop (*Hordeum vulgare*) and its wild relatives (*H. vulgare* ssp. *spontaneum* and *H. murinum* ssp. *glaucum*). The main goal of this research was to study the NB-causing pathogen in the crop center of origin. To address this, we have constructed a *Ptt* (n = 15) and *Ptm* (n = 12) collection isolated from three barley species across Israel. Isolates were characterized genetically and phenotypically. Aggressiveness of the isolates was determined based on necrotrophic growth rate on detached leaves of barley. In addition, isolates were genetically characterized by the mating type, followed by phylogenetic analysis, clustering them into seven groups. The analysis showed no significant differentiation of isolates based on either geographic origin, host of origin or form (*Ptt* vs. *Ptm*). Nevertheless, there was a significant difference in aggressiveness among the isolates regardless of host species, geographic location or sampling site. Moreover, it was apparent that the isolates derived from wild hosts were more variable in their necrotrophic growth rate, compared to isolates sampled from cultivated hosts, thereby suggesting that NB plays a major role in epidemiology at the center of barley origin where most of the diversity lies. *Ptm* has significantly higher necrotrophic and saprotrophic growth rates than *Ptt*, and for both a significant negative correlation was found between light intensity exposure and growth rates.

## 1. Introduction

Net blotch (NB) is a major disease of barley (*Hordeum vulgare*) caused by the necrotrophic fungus *Pyrenophora teres* f. *teres* (*Ptt*) and *P. teres* f. *maculata* (*Ptm*) with associated yield losses ranging between 10% to 40% [1]. The disease was first named after the typical netting symptom produced by *Ptt* on leaves of susceptible barley lines. In addition, a spot-like symptom has been observed and associated with *P*. *teres* f. *maculata* and was defined as *Ptm* a second form of *P. teres*. However, *Ptt* and *Ptm* are indistinguishable morphologically (for example, based on the conidia shape), so their identification can only be based on their symptoms. On a global level *Ptt* and *Ptm* have different global distribution patterns, usually with one form being predominant in a given area [2]. *P*. *teres* forms sexual pseudothecia at the end of the barley growing season. These structures over-summer on infected straw debris which enable pathogen survival between seasons and serve as the main source of inoculation [1,3]. This NB reproductive strategy combines the benefits of sexual reproduction, resulting in new allelic combinations and off-season survival mechanisms, with the advantages of rapidly multiplying individual clonal lines through asexual reproduction [4]. Since *P. teres* propagates very efficiently on wild plants, it forms a huge reservoir of new pathotypes of the pathogen well adapted to wild host populations. This soil-transmitted reservoir could constantly serve as a source of inoculums to initiate epidemics in the fields [1,3]. 

NB is the most economically important disease of barley in Israel. Here, in the center of origin of the crop, it infects not only cultivated barley but also wild barley, e.g., the wild progenitor *H. vulgare* ssp*. spontaneum*, and wild relatives such as *H. murinum* ssp*. glaucum* which are found in natural habitats and as weeds in cultivated fields across Israel. The epidemic burst of NB in 1951, due to the extensive use of NB-sensitive barley varieties and poor crop rotation [3], was the major cause for dramatic and immediate reduction of barley cultivation, formerly a leading crop, in Israel. More than half a century later, with a gradual expansion of barley fields in rain-fed regions in Israel, there is a pressing need to study this pathosystem and identify and dissect durable field resistance against this challenging pathogen [5].

Based on high virulence, high genetic variability of the pathogen and its hosts, and the availability of both reproductive structures (conidia and pseudothecia), Kenneth (1960) concluded that Israel and the surrounding region is a center of pathogen diversity. This could be implying long-term host-pathogen co-evolution in its natural habitats, including a recent phase of co-evolution under domestication. However, it has been illustrated in a few other host-pathogen systems (including barley) that the center of origin of host and pathogen are not necessarily the same [6]. Studying the dynamics of the pathogen and the host at their center of origin, where both natural and domesticated habitats co-exist, could be highly valuable. Recently, several studies have addressed cereal and legume pathosystems in their center of origin. These resulted in the identification of contrasting evolutionary patterns, from pathogen genetic differentiation [7,8,9] to non-differentiated meta-populations [10]. Little is known about the evolutionary processes affecting the NB populations in its presumed center of origin. The close proximity of the four *Hordeum* host species provides a unique opportunity to examine the pathogenetic host range at the host species level. As an examination tool we selected the Mating type gene which controls sexual development and production of sexual spores in *P*. *teres* [11] and was therefore a good candidate for phylogenetic analysis [12] having been already implemented in *P*. *teres* phylogenetic studies. 

The influence of light intensity on the plant-pathogen interaction was studied on several pathogens, such as *Colletotrichum graminicola* on maize and *Botrytis cinerea* on tomato and strawberry. These works showed a negative correlation between light intensity and the ability of the pathogens to grow on leaf tissue and agar plates [10,11]. The barley canopy closure significantly reduces light penetration and induces leaf mortality. In the context of necrotrophic pathogens, light intensity is potentially a major factor that can affect plant-pathogen interactions and the infection process.

In the present study, we report the establishment of a unique ex-situ Israeli collection of NB isolates. We further evaluated NB isolates adaptability using both phenotypic and molecular tools in order to: (i) assign isolates to the *Ptt* and *Ptm* forms; (ii) characterize the eco-geographic range of NB isolates; (iii) characterize the diversity of virulence in the isolates collection; and (vi) test whether the identified variation among isolates could be associated with adaptation to host species, geographic region or environmental constrains such as light intensity.

## 2. Results

### 2.1. Spatial Survey Illustrated that Net Blotch is Distributed Across a Range of Hosts and Eco-Geographical Conditions

A spatial survey of the disease symptoms across Israel, included 126 samples collected from plants exhibiting net blotch (NB) symptoms [2,3] showed that NB disease is distributed across most of the eco-geographical niches of Israel, and infects three out of the four hosts surveyed (few samples of *H. bulbosum* showed symptom, and leaves were collected however *P. teres* could not be isolated). This distribution is sympatric with wild barley species *H. vulgare* ssp. *spontaneum* and *H. murinum* ssp. *glaucum*, and also covers the main area of barley cultivation in Israel (North-West Negev) but excludes *H. bulbosum* (Figure 1, and Appendix A). *H. bulbosum* showed almost no typical NB disease symptoms, and infection carried out in the laboratory (Koch’s postulates) that followed the isolation procedure failed to produce vital spores. Wild *H. vulgare* ssp. *spontaneum* and domesticated barley (*H. vulgare*) were the most common species with NB symptoms, and were found in majority of the sites, based on qualitative assessment (Figure 1, and Appendix A). To enable the identification of patterns in the multivariable eco-geographical data among NB collection sites, we computed clustering analysis based on selected climatic data. This clustering resulted in the classification of sampling sites into six distinguished eco-geographic-clusters, containing few variables (e.g., average solar radiation, precipitation, and temperature between January and March) showing significant differences between these six clusters (Appendix A). 

### 2.2. High Frequency of Pyrenophora teres f. maculata among P. teres Isolates 

The *P. teres* isolates (n = 28) obtained from leaf samples of wild and cultivated hosts from different habitats of Israel, consist a diverse collection of NB isolates (Table 1). The northern and eastern-most isolate was collected from Aloney Habashan (Lat\Long: 33.05\35.83), the southern-most isolate was collected from Gilat (Lat\Long: 31.34\34.67), and the western-most isolate from Kisufim (Lat\Long: 31.38\34.38). The longest distance between two collection sites was 225 kilometers (Table 1, and Appendix A). The NB isolates collection can be divided according to the host species into 19, seven, and two isolates from *H. vulgare* ssp. *spontaneum* (H.S); *H. vulgare* (H.V); and *H. murinum* ssp. *glaucum* (H.G), respectively, or into 21 wild and seven domesticated hosts, respectively. Although samples were collected from plants carrying NB symptoms [2,3], it contained high frequency of *Ptm* (12) and *Ptt* (16). Interestingly, *Ptm* isolates were collected from slightly but significantly warmer sites (average maximum temperature 35.7 ± 0.02 °C, 34.9 ± 0.03 °C for *Ptm* and *Ptt*, respectively, [*P* (|t|) < 0.05] during the off-season, July to September). All other eco-geographic variables from the isolate collection sites did not differ between the *Ptt* and *Ptm* collection sub groups. 

### 2.3. P. teres Isolates Cluster into Several Phylogenetic Groups

To dissect the genetic relatedness of the isolates, we computed a separate Bayesian-based phylogeny [13] for the different Mating type MAT1-1-1 and MAT1-2-1 idiomorph sequences of the sampled isolates. The phylogenetic analysis resulted in identification of three major clades for MAT1-2-1, and four for MAT1-1-1 (based on posterior probability >0.67; Figure 2, and Table 1). No significant differences were found between phylogenetic groups (indicated by numbers adjacent to leaves) and the distribution of *P. teres* f*. teres* or *P. teres* f. *maculate* across the tree; likelihood ratio test [*P* (χ^2^) < 0.184]. In addition, no association was found between phylogenetic groups and host species or eco-geographic-clusters (the latter are indicated with symbols adjacent to the leaves, Figure 2).

### 2.4. Substantial Variability in Saprotrophic Growth Rates 

To further characterize the variation among NB isolates, we examined their saprotrophic growth rate. We observed significant differences in the growth rates of isolates on Czapek Dox agar (Appendix A). We noted that *Ptm* had a significantly higher growth rate (+22%), on average, compared to *Ptt* [0.55 ± 0.02 vs. 0.45 ± 0.04 for *Ptm* and *Ptt*, respectively, *P* (|t|) < 0.033, Table 2]. We did not find any effect of isolate host origin or eco-geographic variables and cluster membership on the saprotrophic growth rate [ANOVA test; *P* (F) < 0.762, *P* (F) < 0.912, respectively]. In addition, no significant differences were found between saprotrophic growth rates of the different phylogenetic groups [Figure 2, Welch’s ANOVA test *P* (F) < 0.45]. 

### 2.5. Necrotrophic Growth Rates Reveal Substantial Variability among Isolates, and Higher Growth Rates of Ptm Isolates

Variability in saprotrophic growth rate was previously associated with the growth rate on leaf tissue [15]. We therefore measured necrotrophic growth rate on barley leaves by inoculating each isolate on four barley varieties ('Noga', 'Ma'anit', 'Sagiv', and 'Barke'). We identified significant differences in necrotrophic growth rate among the fungal isolates (Figure 3, and Appendix A). In line with higher saprotrophic growth of *Ptm*, we found that *Ptm* isolates have significantly higher average necrotrophic growth rate (70%), compared to *Ptt* isolates (0.22 ± 0.02 vs. 0.138 ± 0.05 for *Ptm* and *Ptt*, respectively, *P* (|t|) < 0.01, Figure 3). Differences in necrotrophic growth rates of the NB isolates were not associated with the original host species or the eco-geographic attributes (i.e., eco-geographic-clusters) of the sampling site. No significant differences were found between the phylogenetic groups in distribution of necrotrophic growth rates (ANOVA test and Welch’s ANOVA test *P* (F) < 0.45). 

#### 2.5.1. Isolates from Wild Hosts Have More Variable Necrotrophic Growth Rates

To further characterize the variability in necrotrophic growth rate we compared the effect of wild vs. cultivated host on this phenotype. Isolates derived from wild hosts (H.G and H.S) showed higher virulence variability compared to isolates that were sampled from the domesticated host (H.V). This is demonstrated by the significant difference in the coefficient of variation (CV), i.e., 12.92 vs. 45.95 for domesticated and wild species, respectively [*P* (t) < 0.01, Bartlett test, Table 2]. This variability is also observed by the high representation of isolates from wild hosts on the extreme low (n = 6) and high (n = 9) ends of the virulence scale (Table 2, Appendix A). 

#### 2.5.2. Isolate X Host Interactions

When necrotrophic growth rate was inspected in details for each of the four barley varieties used in the test, some interactions between isolate and host genotype were detected (apparent by crossing of the lines in Figure 3). However, a general trend was apparent from the significant correlation (Spearman's rho > 0.4, (ρ) < 0.0001) between necrotrophic growth rates of NB isolates on each variety-pair (e.g., 0.763 between NB isolates tested on 'Barke' and those tested on 'Ma'anit', Table 3). Generally, 'Barke' had significantly higher sensitivity to NB isolates compared with 'Noga', as expressed by the average necrotrophic growth rate of the isolates, 0.21 ± 0.02 on 'Barke' compared with 0.15 ± 0.01 on 'Noga' (Table 4). This response of 'Barke' to isolates was also characterized by lower variability (although not significant) of isolates necrotrophic growth rates on this host line, compared with 'Noga' (i.e., CV of 47.98 for 'Barke', and 55.14 for 'Noga'). 

#### 2.5.3. Correlation Between Necrotrophic and Saprotrophic Growth Rates among Isolates

We also tested the pathogen phenotype on four barley cultivars (*H. vulgare*) to examine consistency (over different genotypes) and relationship between necrotrophic and saprotrophic growth relationships. We found high and significant correlation [Spearman's rho > 0.4, (ρ) < 0.0001] between necrotrophic and saprotrophic growth rates of the NB isolates on each of the four barley cultivars (e.g., 0.507 for 'Barke', Table 3). Considering this correlation, and the significantly higher saprotrophic and necrotrophic growth rate of *Ptm* compared to *Ptt*, we compared the normalized necrotrophic growth rates of the NB isolates (i.e., necrotrophic growth rate on leaves divided by saprotrophic growth rate) of *Ptt* to those of *Ptm*. The average normalized necrotrophic growth rate of *Ptm* isolates was significantly (18%, *P* (|t|) < 0.05) higher than that of *Ptt*, i.e., 0.39 vs. 0.33, respectively (Table 2). Although this result is in line with the trend shown for necrotrophic growth rates (i.e., higher rate for *Ptm*), the difference between *Ptm* and *Ptt* decreased by approximately 50%, i.e., 18% higher normalized growth rate of *Ptm* over *Ptt* isolates, compared with 70% higher non-normalized necrotrophic growth rate of *Ptm* over *Ptt* isolates.

### 2.6. Light Intensity is Negatively Correlated with Saprotrophic and Necrotrophic Growth Rates

The NB habitat in cultivated barley fields and the natural wild habitats of the *Hordeum* wild species that were sampled are characterized by differing vegetation densities and corresponding light intensities levels. We therefore tested the effect of light intensity on saprotrophic and necrotrophic growth rates across all isolates and the four cultivars used in the assays. Significant negative polynomic correlation were found between the intensity of light exposure and the saprotrophic and necrotrophic (R^2^ = 0.99 and 0.92 respectively) growth rates (Figure 4), reflecting a diminishing effect at the higher light intensity values. In all assays, the fastest growth occurred under the lowest light intensity (Figure 4, Supplementary). 

## 3. Discussion

Barley was one of the earliest crops to be domesticated together with other major cereal crops [16] in the Fertile Crescent. Since then, the NB pathosystem transformed from merely a wild based pathosystem to a mixed co-existence of wild habitat and cultivated barley fields distributed sympatrically. The present study aimed to characterize a major barley pathogen, the net blotch causal agent, in the center of origin of the crop, to construct a representative collection of this pathogen, and to characterize its genetic variability and virulence profile. In addition, we discuss the relative contribution of the *Ptt* and *Ptm* genetic forms to the NB epidemics in our region. 

### 3.1. Disease Distribution

In Israel and surrounding regions, wild barley species thrives across a wide ecological amplitude often sympatric to agricultural barley fields [17,18]. Moreover, both *H. vulgare* ssp. *spontaneum* and *H*. *murinum* ssp. *glaucum* are ruderal species which are found as weeds in high intimacy with the cultivated crop. 

In general, NB propagates efficiently on wild plants (Figure 1, and Appendix A). Moreover, there is a huge reservoir of the pathogen in the wild host populations, which could occasionally or continuously serve as a source of initial inoculum to epidemics in the field [5,19]. Based on the frequency of infected leaf samples in our survey, we can conclude that wild *H. murinum* ssp. *glaucum*, and *H.* vulgare ssp. *spontaneum* and domesticated *H.* vulgare are the main hosts of *Ptt* and *Ptm* in our region, as reported in previous studies [3], and therefore serve as the major inoculum reservoir. Our observations also suggest that the best time to find clear NB symptoms is during the month before heading (when canopy is still green and before other pathogens rise), as previously suggested [3,5]. *H. bulbosum* showed almost no symptoms of either *P. teres* form, and even when there were symptoms, we did not find vital conidia. The fact that no successful isolation occurred is in full accordance with Kenneth (1960) finding and support his assumption that *H. bulbosum* is a NB non-host. This result supports the suggestion that *H. bulbosum* should be considered as a good resource of NB resistance [20,21,22].

The environment is considered a fundamental rib in the epidemic classical triangle [23]. We have looked into a long list of climatic variables, and identified significant differences between sampling sites for a few variables (Appendix A) that may affect NB epidemics and host physiology. However, no association was observed between isolate saprotrophic and necrotrophic growth rates and the eco-geographical conditions in their site of origin, as reflected by the eco-geographic clusters we identified (Appendix A). Consistently, we did not find association between the host species from which the isolate was sampled and the environment (eco-geographic clusters). Our findings are in line with a similar study in the same region on the legume pathogen *Peyronellaea pinodes* which did not identify association between mean temperature in the origin of the pea isolate and temperature responses, spore germination rate, and aggressiveness [10]. 

### 3.2. Characterization of the NB Isolates Collection

#### 3.2.1. Genetic Characterization

In this study, we computed a Bayesian-based phylogeny of isolates from the center of origin of barley, the host of *P. teres*. Phylogenetic analysis enabled us to identify several phylogenetic clusters, partially associated to host species. However, we did not identify association between the phylogenetic groups (based on the MAT1 locus) and the environmental parameters (eco-geographic-clusters) that were tested, nor with the necrotrophic and saprotrophic growth rates, and neither with the classification into the two forms of *P. teres*. This is in contrast to Rau et al. [24] who report on correlation between latitude of collection site and necrotic growth. The lack of correlation with environmental conditions may suggests that the mating type gene used in our study to reconstruct the phylogenetic history of *P. teres* is not significantly linked to mechanisms that support adaptation to these conditions. The lack of correlation to *Ptt* and *Ptm* is in contrast to previous studies that illustrated, based on MAT1, distinct clusters of the two forms [25,26]. Additionally, Rau et al. [24] reported on different co-evolutionary strategies of *Ptt* and *Ptm* on a particular barley landrace across six sites in the Island of Sardinia. Our research, which summarized the data of 28 isolates from both wild and domesticated barley hosts at the center of origin of this pathosystem, did not provide evidence for such *Ptt* and *Ptm* differentiation. These differences may result from the different sets of isolates and different algorithm used to reconstruct phylogeny (distance matrix-based methods) in the previous studies. As Bayesian inference phylogeny [13] is based on multiple sequence alignment, it is focused on conserved evolutionary information presumably important for the function and structure of the protein coding gene. Considering the visual-based classification (into *Ptt* and *Ptm*) some differences between studies may occur, however the significantly higher pathogenicity, and saprotrophic growth rate observed for the *Ptm* isolates furthers supports classifying them into one group. We have also tested the potential use of ITS sequences to construct a phylogenetic tree. Although ITS serves as a good “barcode” for differentiating fungal species [14], it did not contain sufficient variability to generate genetic resolution between isolates of *P. teres* (i.e., isolates of the same specie), as previously demonstrated by Andrie et al. [27]. 

#### 3.2.2. Characterization of Virulence on Wild vs. Domesticated Host

Analysis of necrotrophic and saprotrophic growth rates showed substantial variance among the isolates, with significant effect of the host genotype (different barley cultivars used for the assay, Figure 3, Table 2). No differences in aggressiveness were found between isolates originating from wild vs. domesticated hosts. Similar to our study, no association was found by Golani et al. [10] between isolates from domesticated or wild pea in temperature responses, spore germination rates, and aggressiveness. This lack of differentiation in the NB pathosystem may be a result of the ruderal character of wild barley which could prevent specialization of *P. teres* into two different groups, that attack either wild or cultivated barley, since these species are growing side by side, providing an advantage to less specialized isolates. Such intimacy between wild and domesticated hosts is rarely observed in wheat, therefore enabling the slow and co-evolving differentiations of the pathogens in the wild habitats compared to domesticated field [7,8]. However, this conclusion should be carefully taken due to the relatively small sample size of isolates sampled from domesticated host.

Interestingly, the variability in virulence between isolates was associated with host origin (wild vs. cultivated), as expressed by higher CV values (for necrotrophic growth rate) in isolates originating from the wild (Table 2). However, these two groups did not differ in mean virulence values. The higher CV values may be explained by the higher genetic variability of wild barley compared the domesticated barley [28]. An additional cause might be the widespread distribution of wild barley in Israel, which causes high variability within and between populations on the wild host due to changing environmental conditions (expressed across our six eco-geographic clusters), ultimately increasing variability of the pathogen population [29]. Previous studies reported variability in pathogenicity between host species. Isolates of *Blumeria graminis tritici* (*Bgt*) in wheat showed differences in aggressiveness between the wild and the domesticated groups of isolates [7]. Similarly, Ben-David et al. [8] found higher CV values in virulence profiles of *Bgt* isolates from wild origin, compared to *Bgt* collected from wheat fields. 

Studies which used differential lines already showed different R and susceptibility genes [30] for NB in barley. However, our assays of necrotrophic growth rate were based on host varieties from farmer fields, lacking data regarding their genetic background or presence of resistance genes. Therefore, our conclusions are mostly general and cannot shed light on the actual genetic mechanism that governs host plant response. 

#### 3.2.3. The Prominent Role of *P. teres* f. *maculata*

This work highlights the role of *P. teres* f. *maculata* in the NB pathosystem. First, the NB spatial survey demonstrated high frequency of this form encompassing approximately 40% of the samples. Interestingly, this high frequency of *Ptm* was not reported in the last NB survey, done in Israel in 1960 [3]. However, the dominant role of *Ptm* was recorded lately in a survey of barley fields in Victoria, Australia [31]. Although our survey is confined to 28 samples, our findings demonstrate a higher virulence of *Ptm*, as shown by 70% and 20% higher necrotrophic and saprotrophic growth rates, respectively, compared with *Ptt* (Table 2). These finding is in full accordance with the study on NB in Sardinia [24]. *Ptm* had also significant (20%) higher normalized virulence (measured by necrotrophic divided by saprotrophic growth rate) suggesting that the difference in virulence is not only related to the higher saprotrophic growth rate, and could result from additional virulence mechanisms (e.g., fungal effectors; [15]). The contribution of host defense mechanisms to this result is probably minor, considering that higher virulence of *Ptm* was observed on all four tested host lines (Figure 3). *Ptm* was found in sites with significantly higher maximum temperature in the warm off-season (from July to September), which could explain the recent increase in *Ptm* abundance in new area due to the evidence of rise in temperature [2,32,33]. We are aware that an increase of nearly 1 ^o^C is not likely to affect fungal survival, however such a mean value is based on 30 years data, and may therefore reflect long-term climatic changes in this eco-geographic region. This change could have affected the pathogen over-summering adaptation and survival through a presumably unknown mechanism. Finally, our phenotypic assays and analysis of climatic data from the eco-geographic survey, illustrate that *Ptm* isolates exhibit higher aggressiveness and adaptation to higher temperatures, respectively. *Ptm* may therefore play an important role in a future scenario of increasing temperatures (i.e., climate change), thus the control of this form of *P. teres* should be considered in future breeding programs [31,34]. Our phylogenetic analysis showed significant genetic variability (illustrated by distinct clades). In addition, our analyses of saprotrophic, necrotrophic and light dependent growth show significant substantial differences between isolates. For example, our analysis illustrates approximately a 3.5 fold difference between the highest and lowest saprotrophic growth rates (Appendix A), similarly the difference between the highest and lowest necrotrophic growth rates on ‘Barke’ is 7.1-fold (bigger differences were illustrated on the three other cultivars). Thus, it is conceivable that this study is likely to represent major trends of diversity of *Ptt* and *Ptm* isolates.

#### 3.2.4. Reduced Light Intensity, a Potential Driver of NB Epidemics

We have detected a negative correlation between light intensity and pathogen saprotrophic and necrotrophic growth rates. NB growth rate is maximal at low light intensity (Figure 4). The significant role of light intensity on necrotrophic and saprotrophic growth rates of the NB isolates is consistent with previous studies of another necrotroph, *Botrytis cinerea* [35,36]. An initial study showed approximately 70% increase in saprotrophic growth rate, and 400% increase in lesion size on *Arabidopsis thaliana* leaves, when *B. cinerea* was grown in constant darkness compared with constant light [35]. Subsequent study demonstrated the presence of a circadian oscillator in *B. cinerea* and its role in pathogenicity; lesions produced by *B. cinerea* on *A. thaliana* leaves were smaller at dawn compared with dusk, and this effect depended on the pathogen's circadian system, as it was abrogated by over expression of the negative regulator BcFRQ1, or by constant light [37]. The role of light intensity in pathogenicity is an emerging topic, demonstrated so far in only a few systems [37]. Our work illustrates this phenomenon in an important pathogen of a staple crop, based on monitoring 108 different isolate X host interactions (27 isolates x four barley cultivars), providing analytic and statistical robustness to our finding. The shape of cereal canopy (in general), and particularly in barley, reduces the light intensity. The vegetative growth stage of cultivated barley is often characterized by a 'green-cap', made of young leaves covering and over-shading older leaves. Considering that important part of the disease cycle of the *Ptt* and *Ptm* is occurring under this green cap, fungal isolates that evolved to have a faster necrotrophic growth rate under low light intensity would have a competitive advantage on other isolates and improved adaptively to the host.

## 4. Materials and Methods

### 4.1. Sampling Procedure

A total of 126 samples of barley leaves with NB symptoms from four host species (*H. vulgare* ssp. *spontaneum*, H.S; *H. vulgare*, H.V; *H. murinum* ssp. *glaucum*, H.G; and *H. bulbosum*, H.B) were collected during 2015–2017 across Israel. Sampling was done from a wide range of eco-geographic environments (Figure 1, Appendix A). Collection site latitude, longitude and elevation were measured by GPS. Long-term data on precipitation (annual average, amount in the wet month), maximum and minimum temperatures, wind speed average and solar radiation were extracted from the 30 years WORLDCLIM dataset (~1 km spatial resolution) [34]. The choice of which variable to include in the eco-geographic clustering analysis was based initially on their presumed influence on NB dynamic and epidemics at the field during the season. This was done based on our experience from the current survey and previous work [3]. The dried leaves samples were kept in the dark at room temperature prior to fungal isolation. These conditions maintain the vitality of the isolate cultures for more than one year.

### 4.2. Plant and Pathogen Material

Twenty-seven isolates of *Ptt* and *Ptm* were isolated from the leaf samples, representing the barley host and eco-geographic range (Appendix A). A modified protocol [3,38] was used to develop single-conidial culture from natural infected leaves. In short, leaf samples were cut into 20–30 mm pieces, surface sterilized by 70% alcohol for 10 s and washed in distilled water. Each sample was placed in a clean petri dish containing a wet filter paper (Whatman) and incubated at 18–20 °C with 12 h light/12 h dark for 36 hrs. Single conidia were transferred (under light microscope) to PDA (potato dextrose agar) plates, by Koch's postulates on detached leaves [3]. Species validation level was based both on alignment of the MAT1 gene (below) of isolates sequences to the GenBank database (NCBI) using BLASTN, and on plant symptoms. The classification into *Ptt* or *Ptm* were done using the symptom appearance as described in the literature [1,2,39].

### 4.3. Genomic DNA Extraction, Primers, PCR, and Sequence Analysis

DNA extraction was performed according to Möller [40] from fresh mycelia in liquid Czapek Dox medium with 0.25 g/L chloramphenicol (Sigma). PCR amplification of the MAT1 gene (primers in Table 5) was conducted as described by Lu et al. (2013), followed by sequencing of the product by Sanger sequencing at Hy Laboratories Ltd, Rehovot, Israel.

### 4.4. Phylogenetic Analysis 

The MAT1 gene sequences were processed by Geneious® software (https://www.geneious.com); nucleotides with sequencing quality below 40 were excluded from the analysis [Quality < 40, *P* (error) < 0.0001]. A multiple sequences alignment (MSA) was generated separately for isolate sequences of MAT1-1-1 or MAT1-2-1 idiomorphs, using MAFFT (global-pair algorithm) [41]. Gaps were removed from each MSA using trimAL (auotomated1 algorithm) [39]. The best model was estimated using the IQ-TREE built-in function, which tests 144 models (Minh et al. 2013). Bayesian inference was computed using MrBayes, under the GTR substitution model with an invariable gamma distribution for 300,000 generations for both mating type genes [13]. FigTree [42] was used to generate phylogenetic trees. The isolates Timrat and Gan-Yavne (Appendix A), which appear in virulence characterization tests, do not appear in this analysis due to difficulty in extracting DNA and obtaining informative sequence from these isolates. We also attempted to identify differences between the isolates based on other “housekeeping” genes such the Internal transcribed spacer (ITS) sequence [43], however their sequences did not contain enough variability to differentiate between isolates. As MAT1 contains highest sequence variability among *Ptt* and *Ptm* isolates it is more likely to separate between these different forms.

### 4.5. Phenotypic Assay

Saprotrophic growth rate. The saprotrophic growth rate was tested as follows: An agar plug (4 mm diameter) containing mycelium of the tested isolate (taken from the margins of the growing colony) was placed on Czapek Dox agar with 0.25 g/L chloramphenicol (10 ml per each plate). Each isolate was tested in four repeats, and four plates per isolate were arranged in a stack (as described in the detached-leaf experiment). The experiment was performed in two rounds: (1) 26 isolates were represented by four repeats, each exposed to a different light intensity. Repeats 1-4 were exposed to 1800-2800, 320-680, 90-180, and 45-120 lux, respectively; (2) 16 selected isolates were tested in three different light intensities: 3500, 1500, and 500 lux, respectively. Growth on either detached leaves (see below) or Czapek Dox medium was photographed using a Canon 100D (30-2, Shimomaruko 3-chome, Ohta-ku, Tokyo 146-8501, Japan), once or twice a day. The images were analyzed using ImageJ software to calculate the rate of progress of each isolate [44]. Isolate ‘Beeri’ was omitted from the saprotrophic and necrotrophic growth analysis as it ceased growing during the course of the study.

Necrotrophic growth rate. The necrotrophic growth rate was measured on detached barley leaves of cultivated barley lines ‘Noga’, ‘Ma’anit’, ‘Sagiv’, and ‘Barke’. The first three are commercial cultivars presently grown in Israel, the last is a German variety. The first leaf of each seedling was cut into seven cm segments and placed together in 9 × 9 cm^2^ high density polyethylene boxes on sterilized four-layer paper towel. In each box six ml of sterile water containing 60 ppm of benzimidazole (Sigma) were added [38]. Each detached leaf was inoculated in the center with an agar plug (four mm diameter) of mycelium from the margins of a growing colony of a given isolate. Each isolate was tested on all four varieties in four replicates. During the first 24 h the plates were placed in a biological hood at 24 °C and 400 lux, then they were transferred to an incubator at 18 °C. The boxes of each isolate were arranged in a stack, thus the upper replicate was exposed to the highest light intensity and the bottom one to the lowest intensity (3500, 1280, 500, and 280 lux for each replicate, respectively). The terms aggressiveness and virulence are defined as the quantitative negative effect of a pathogen on its host and the capacity of the pathogen to infect a particular host genotype, respectively, following Vanderplank terminology [29].

### 4.6. Statistical Analyses

All statistical analyses were performed on JMP® 13 (SAS Institute Inc., Cary, NC, 1989–2007). Testing for a normal distribution was carried out by the Shapiro-wilk W test [*P* < 0.05] and homogeneity of variances was estimated using the Bartlett test. ANOVA was applied to assess the effect of different factors (host species, eco-geographical variables, the fungus *Ptt* and *Ptm*, genetic clade according to phylogenetic analysis, and light intensity) on necrotrophic and saprotrophic growth rates. ANOVA was also used to test whether isolates groups, divided according to factors such as host, fungus form (*Ptt* or *Ptm*) and genetic clade (phylogenetic analysis) showed significant differences in eco-geographic variables from their site of collection. In cases of non-normal distribution, we used nonparametric tests: Kruskal–Wallis test (instead of one-way ANOVA) and Wilcoxon signed-rank test (instead of Student's t-test). Ward's method was used to cluster the samples based on their eco-geographic variables data.

## 5. Conclusions

The NB isolate collection reported here is a promising resource for studying plant-pathogen interaction in the center of host diversity. In this work, we have shown that *Ptm* has a prominent representation in the pathogen population with a clear advantage over *Ptt* in necrotic growth. The relative dominance of *Ptm* in the NB-Barley pathosystem at the center of origin was not documented before and might reflect a long-term adaptation to climate change. This result, together with a presumable better adaptation of *Ptm* based on its higher growth rate, highlights the importance of this form in field epidemics, suggesting that future studies and breeding programs should focus on the control of this pathogen. Additionally, the variability in pathogenicity of the isolates may lead to the conclusion that several representative isolates should be used in breeding programs aimed to develop durable resistance. Such programs should consider the interaction between the barley varieties and the isolates as illustrated in our study. Finally, a significant negative correlation was found between light intensity and isolate virulence. In agreement with previous studies on another necrotroph [35,37], this result suggests that light sensing could play an important role in the disease cycle of the NB-Barley pathosystem. 

## Figures and Tables

**Figure 1 pathogens-08-00275-f001:**
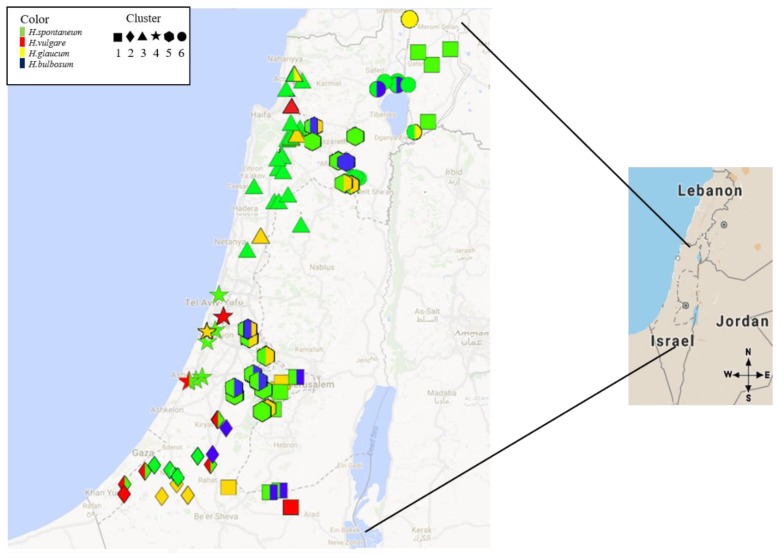
Spatial survey of *Pyrenophora teres* samples (that exhibited net blotch (NB) symptoms) from wild barley (*H. spontaneum, H. murinum* ssp. *glaucum*, *H. bulbosum*) or cultivated barley (*H. vulgare*) fields (host species are indicated by colour code) from different eco-geographic regions (co-geographic-clusters are indicated by symbol).

**Figure 2 pathogens-08-00275-f002:**
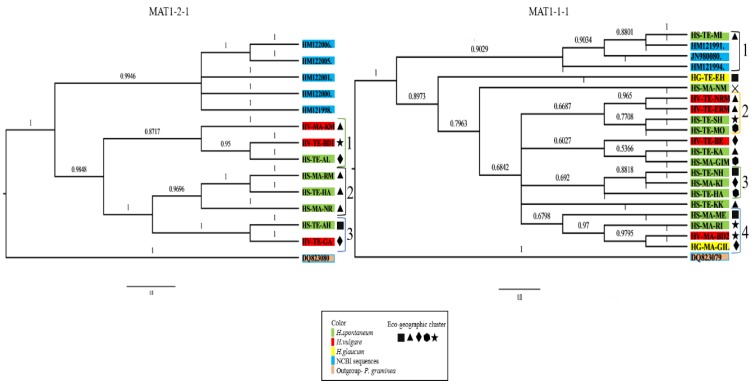
Phylogeny computed by Bayesian inference, based on the full-length of MAT1-2-1 (left) and MAT1-1-1 (right) idiomorphs. Three major MAT1-2-1, and four major MAT1-1-1 clades are indicated by numbers right to the leaves. Bayesian posterior probability values are shown above the branches. Wild and cultivated hosts (*H. vulgare* ssp. *spontaneum,H. murinum* ssp. *glaucum*, *H. bulbosum*, and *H. vulgare*) from which isolates originated are indicated by color code, and the eco-geographic classification (eco-geographic clusters) of each host's environment is indicated by symbol. Another species of this genus, *P. graminea**,* was used as an outgroup [14], and sequences of non-Israeli isolates (NCBI sequence) were used as a reference.

**Figure 3 pathogens-08-00275-f003:**
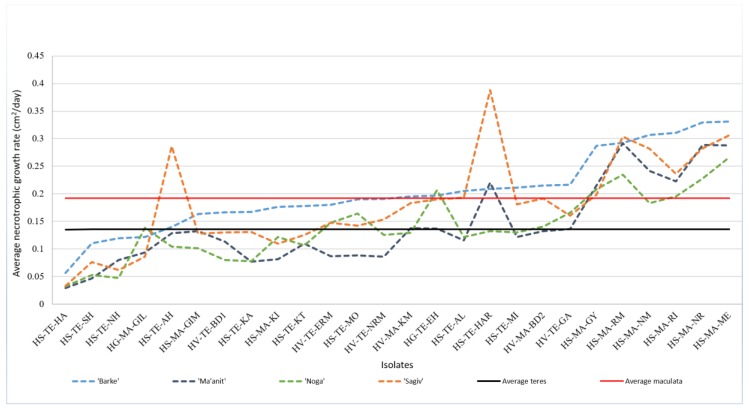
Comparison of mean necrotrophic growth rate (cm^2^\day) of each of the 27 NB isolates (Table 1), on detached leaves of each of four barley varieties separately (four replicates of detached leaves from each variety were used for each isolate). The mean necrotrophic growth rate of *Ptm* and *Ptt* are illustrated in red and black lines, respectively.

**Figure 4 pathogens-08-00275-f004:**
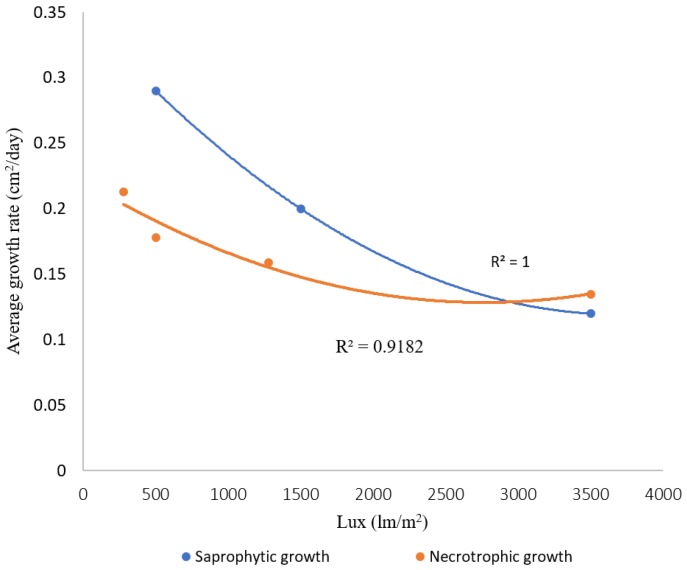
Correlation between light intensity (Lux) and average saprotrophic or necrotrophic growth rates (cm^2^/day) of 27 NB isolates, tested on four different barely varieties ('Noga', 'Ma'anit', 'Sagiv', and 'Barke'). Significant regression was found for saprotrophic (*P* (F) < 0.0018), and necrotrophic (*P* (F) < 0.0001) growth rates, respectively.

**Table 1 pathogens-08-00275-t001:** Characterized *Ptt* and *Ptm* isolates used in this study.

Isolate Name *^1^*	Collection Site	Host Specie	MAT Idiomorph	Forma	Phylogenetic Group
H.S-MA-RI	Rishon	*H. spontaneum*	1-1-1	*Ptm*	4
H.V-TE-NRM	Ramot Menashe	*H. vulgare*	1-1-1	*Ptt*	2
H.V-TE-ERM	Ramot Menashe	*H. vulgare*	1-1-1	*Ptt*	2
H.V-TE-BE *^2^*	Beeri	*H. vulgare*	1-1-1	*Ptt*	-
H.V-MA-BD2	Beit Dagan	*H. vulgare*	1-1-1	*Ptm*	4
H.S-TE-SH	Shtulim	*H. spontaneum*	1-1-1	*Ptt*	2
H.S-MA-NM	Neve Michae	*H. spontaneum*	1-1-1	*Ptm*	-
H.S-TE-NH	Netiv Halamed He	*H. spontaneum*	1-1-1	*Ptt*	3
H.S-TE-MO	Modi'in	*H. spontaneum*	1-1-1	*Ptt*	2
H.S-TE-MI	Mitzpe Ilan	*H. spontaneum*	1-1-1	*Ptt*	1
H.S-MA-ME	Meytzar	*H. spontaneum*	1-1-1	*Ptm*	4
H.S-MA-KI*^3^*	Kisufim	*H. spontaneum*	1-1-1	*Ptm*	3
H.S-TE-KA	Katzir	*H. spontaneum*	1-1-1	*Ptt*	-
H.S-TE-KT	Kiryat Tivon	*H. spontaneum*	1-1-1	*Ptt*	-
H.S-TE-HAR^3^	Haruvit	*H. spontaneum*	1-1-1	*Ptt*	3
H.S-MA-GIM	Gimzue	*H. spontaneum*	1-1-1	*Ptm*	-
H.G-MA-GIL	Gilat	*H. glaucum*	1-1-1	*Ptm*	4
H.G-TE-EH	El Hahar	*H. glaucum*	1-1-1	*Ptt*	-
H.S-TE-AH	Aloney habashan	*H. spontaneum*	1-2-1	*Ptt*	3
H.V-MA-KM *^3^*	Kefar Masarik	*H. vulgare*	1-2-1	*Ptm*	1
H.V-TE-GA	Gat	*H. vulgare*	1-2-1	*Ptt*	3
H.V-TE-BD1	Beit Dagan	*H. vulgare*	1-2-1	*Ptt*	1
H.S-MA-RM	Ramot Menashe	*H. spontaneum*	1-2-1	*Ptm*	2
H.S-MA-NR	Nahal Raz	*H. spontaneum*	1-2-1	*Ptm*	2
H.S-TE-HA	Harish	*H. spontaneum*	1-2-1	*Ptt*	2
H.S-TE-AL	Alumim	*H. spontaneum*	1-2-1	*Ptt*	1
H.S-MA-GY	Gan Yavne	*H. spontaneum*	n/a	*Ptm*	n/a
H.S-MA-TI *^3^*	Timrat	*H. spontaneum*	n/a	*Ptm*	n/a

1: Isolates names sections, separated by hyphen, correspond to host-form-location, e.g., H.S-MA-RI corresponded with *H. spontaneum*- *Ptm*-Rishon; 2: This isolate was omitted from the saprotrophic and necrotrophic growth analysis as it ceased growing in the course of the study; 3: Isolates that all three primers were used due to low sequences quality.

**Table 2 pathogens-08-00275-t002:** Coefficient of variation (CV) and mean of necrotrophic (cm^2^/day) and saprotrophic [cm(diameter)/day] growth rates on detached leaves, and on Czapek Dox agar, respectively.

			No. of Isolates	CV			Mean^4^		
				Nec. Growth Rate *^1^*	Sap. Growth Rate *^2^*	Nec. / Sap. *^3^*	Nec. Growth Rate	Sap. Growth Rate	Nec. / Sap.
Category	forma	*Ptm*	12	15.09	34.76	26. 53	0.22 ^A,*^	0.55 ^A,*^	0.39 ^A,*^
		*Ptt*	15	34.13	34.48	42.41	0.13 ^B^	0.45 ^B^	0.33 ^B^
	Host	Domesticated	6	12.92 ^A,**^	26.93	37.77	0.15	0.48	0.36
		Wild	21	45.95 ^B^	28.28	42.09	0.17	0.50	0.35

*^1^* Necrotrophic growth rate on detached leaves. *^2^* Saprotrophic growth rate on Czapek Dox agar. *^3^* Necrotrophic/Saprotrophic growth ratio. *^4^* Mean comparison is preformed within each category between different row pairs of the same column (e.g., average necrotrophic growth rate was significantly different between *Ptm* and *Ptt* in the forma category). Different letter mark means which are significantly different. ^*, **^ indicate significant of Student's *t*-test (*P* (|t|) < 0.05) and Bartlett test (*P* (*t*) < 0.01), respectively.

**Table 3 pathogens-08-00275-t003:** Correlation [Spearman's rho (ρ)] of necrotrophic growth rates of NB isolates between variety-pairs, and between necrotrophic and saprotrophic growth rates of a given variety.

	Variety, medium				
Variety, medium*^1^*		Noga''	'Ma'anit'	'Sagiv'	'Barke'
	‘Noga’				
	'Ma'anit'	0.682*			
	'Sagiv'	0.684*	0.761*		
	'Barke'	0.760*	0.763*	0.740*	
	'Czapek'	0.424*	0.448*	0.405*	0.507*

*^1^* Variety refer to 'Noga','Ma'anit', 'Sagiv', and 'barke', and medium to 'Czapek'; * Significant value *P* (ρ) < 0.0001.

**Table 4 pathogens-08-00275-t004:** Coefficient of variation and mean necrotrophic growth rates (detached leaf assay) of NB isolates on the four barley -cultivars.

Variety	CV	Average *^1^*	N
'Barke'	47.989	0.211^A^	105
'Sagiv'	55.479	0.18^AB^	105
'Ma'anit'	61.267	0.15^AB^	105
‘Noga’	55.146	0.145^B^	108

*^1^* Data refer to the average of four repeats. Different letter mark means which are significantly different. by Tukey HSD test (*P* < 0.05).

**Table 5 pathogens-08-00275-t005:** Oligonucleotide primers used in this study.

Primer Name	Sequence (5′→3′)	Amplicon Size (In Base Pairs)
*ptt*-MAT1F*^1^*	TGGAAGGATCGCAGACTGGAA	1933-2161
*ptt-*MAT1R*^1^*	TTCGTCGCGGAGGAGGCTTGT	
*ptt-*MAT1-1F495*^2^*	GGCAACAAGAGGTGAAGGTG	1273

*^1^* The primers were suitable for both MAT1-1-1 and MAT 1-2-1 as well [26]. *^2^* Used on isolates with low quality of sequence, with *ptt*-MAT1R.

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
