# Peer review of "Characterization of the Barley Net Blotch Pathosystem at the Center of Origin of Host and Pathogen"

_pathogens, 2019, doi:10.3390/pathogens8040275_

Round 1

Reviewer 1 Report

In this publication, the authors have sampled 126 barley plants with Net Blotch (NB) symptoms from four different host species across a wide eco-geographic are in Israel. The authors then perform correlation analysis from each sample with climate data, saprotrophic and necrotrophic growth rate, light intensity and phylogeny based on the MAT1-1-1 and MAT1-2-1.

However, most of these correlations yield negative results, such as there seems to be no correlation between the host species and the climate data, no association between phylogeny and environmental conditions and growth rates, no significant differences of virulence between hosts, etc. 

As such the main conclusions of the study are that there are differences levels of virulence between the pathogen isolates, higher prevalence and virulence of Ptm within the samples collected and a negative correlation between light intensity and pathogen growth rates. In order to improve the impact of these findings a better understanding of the genetic background of both the host and the pathogens could be obtained by using genotyping technologies. For the light intensity, the conclusion is based on in-vitro experiments and extrapolating conclusion to the field should be taken carefully. If this claim is true what could explain the shouldn't there be a correlation between virulence, disease spread and solar radiation? Why isn't this the case? 

Finally, the introduction could be improved by providing the rationale to some of the decision made. Why were these collections samples chosen, why was phylogeny done only on MAT genes, how do these findings correlate to the world diversity of Ptt and Ptm? 

Reviewer 2 Report

Very interesting work, time-consuming and labor-intensive research. the reviewer only has reservations about the presentation of the results. Figure 3 in particular is illegible.
Material and methods:
In the case of sampling, please specify whether the same samples were taken for 3 years and the results are presented as average?
If not, please separate the test material and indicate which were analyzed in individual years. then please do a statistical analysis and compare them for years. Weather conditions have a significant impact on the occurrence of pathogens.

Round 2

Reviewer 1 Report

I want to thank the authors for the clarification regarding my points on the previous review and I am happy with the changes introduced to clarify some of the rationales behind their decisions. Therefore I would recommend the article for publication